CellPress

## Commentary

# International federation of genomic medicine databases using GA4GH standards

Adrian Thorogood,[1,2,*] Heidi L. Rehm,[3,4] Peter Goodhand,[5,6] Angela J.H. Page,[4,5] Yann Joly,[2] Michael Baudis,[7] Jordi Rambla,[8,9] Arcadi Navarro,[8,10,11,12] Tommi H. Nyronen,[13,14] Mikael Linden,[13,14] Edward S. Dove,[15] Marc Fiume,[16] Michael Brudno,[17] Melissa S. Cline,[18] and Ewan Birney[19]

[1]ELIXIR-Luxembourg and Biocore, Luxembourg Centre for Systems Biomedicine, University of Luxembourg, Belvaux, Luxembourg
[2]Centre of Genomics and Policy, Department of Human Genetics, McGill University, Montreal, QC, Canada
[3]Center for Genomic Medicine, Massachusetts General Hospital, Boston, MA, USA
[4]Broad Institute of MIT and Harvard, Cambridge, MA, USA
[5]Global Alliance for Genomics and Health, Toronto, ON, Canada
[6]Ontario Institute for Cancer Research, Toronto, ON, Canada
[7]University of Zurich and Swiss Institute of Bioinformatics, Zurich, Switzerland
[8]Centre for Genomic Regulation (CRG), The Barcelona Institute of Science and Technology, Barcelona, Spain
[9]Universitat Pompeu Fabra, Barcelona, Spain
[10]Institute of Evolutionary Biology (UPF-CSIC), Department of Experimental and Health Sciences, Universitat Pompeu Fabra, Barcelona, Spain
[11]Institució Catalana de Recerca i Estudis Avançats (ICREA), Barcelona, Spain
[12]Barcelonaβeta Brain Research Center (BBRC), Pasqual Maragall Foundation, Barcelona, Spain
[13]CSC - IT Center for Science, Life Science Center, Espoo, Finland
[14]ELIXIR-Europe (Finland), Wellcome Genome Campus, Hinxton, Cambridgeshire, UK
[15]School of Law, University of Edinburgh, Edinburgh, UK
[16]DNAstack, Toronto, ON, Canada
[17]Department of Computer Science, University of Toronto and University Health Network, Toronto, ON, Canada
[18]UC Santa Cruz Genomics Institute, Mail Stop: Genomics, 1156 High Street, Santa Cruz, CA 95064, USA
[19]European Molecular Biology Laboratory, European Bioinformatics Institute, Wellcome Genome Campus, Cambridgeshire, UK
*Correspondence: adrian.thorogood@uni.lu

We promote a shared vision and guide for how and when to federate genomic and health-related data sharing, enabling connections and insights across independent, secure databases. The GA4GH encourages a federated approach wherein data providers have the mandate and resources to share, but where data cannot move for legal or technical reasons. We recommend a federated approach to connect national genomics initiatives into a global network and precision medicine resource.

## Introduction

National-scale genomic sequencing initiatives are emerging worldwide to promote personalized healthcare and innovation. These national initiatives will generate genomic datasets for tens of millions of individual people as part of routine healthcare.[1] Connecting this wealth of data internationally offers great potential to advance our understanding of and our ability to address disease. Genomic and health-related data are sensitive, however, implicating the privacy of sequenced individuals and their families and typically attracting legal restrictions on disclosure and potentially also international transfer. The Global Alliance for Genomics and Health (GA4GH) is a standards-setting body established to promote the international sharing of genomic and health-related data.[1] It supports diverse models for sharing genomic and health-related data with authorized users while also protecting competing interests. These models span central databases to networks of distributed databases connected by common infrastructure.[2] Data can be hosted in the cloud—along with methods, workflows, and computing resources—to facilitate secure, international access and large-scale analysis.[3]

A federated approach to data sharing is an alternative in which independent data providers maintain their own secure database. A data provider is any organization hosting a database of genomic and associated health data willing to share the data with data users—individuals and organizations who seek to analyze data. By adopting data and technical standards, they enable users to analyze data across multiple databases and combine the results. Each data provider maintains full control over its data and access management in a secure computing environment. Data providers may choose to voluntarily align on common access policies and infrastructure to streamline user experience (Figure 1).[4,5] Federated approaches are highly attractive in principle, offering data providers more control without sacrificing opportunities for collaboration and openness. The concept is also flexible and can be adapted to different contexts. This flexibility can, however, lead to disagreement over what federated data sharing means in practice, stymying implementation.

In this commentary, we promote a shared vision for how and when to federate genomic and health-related databases. We review central considerations for developing these federated systems, including key design choices and trade-offs, and

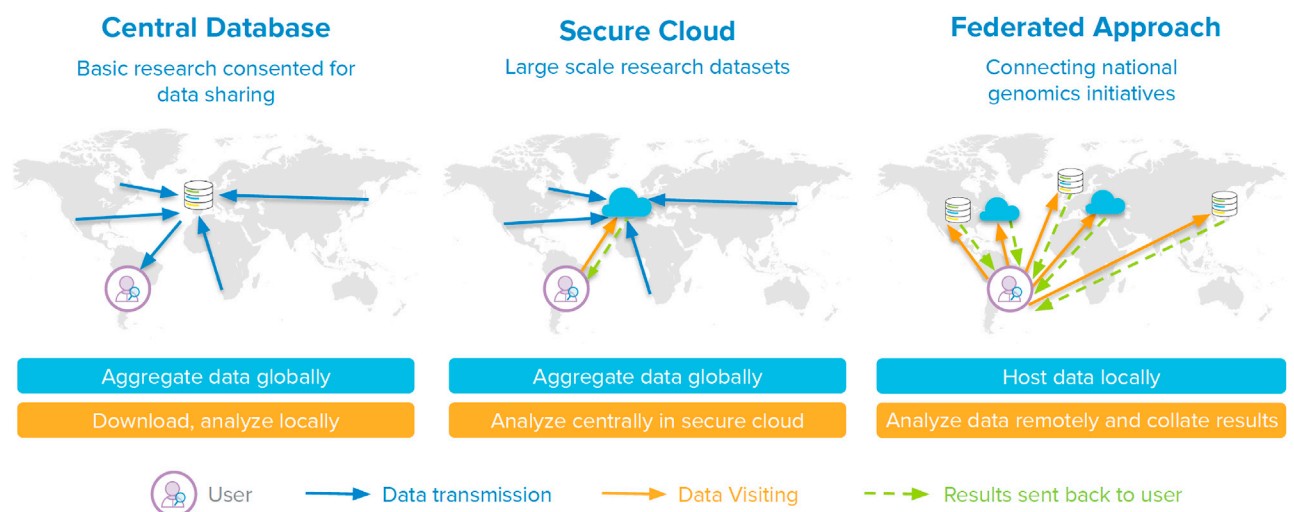

**Central Database**
Basic research consented for data sharing

**Secure Cloud**
Large scale research datasets

**Federated Approach**
Connecting national genomics initiatives

Aggregate data globally
Download, analyze locally

Aggregate data globally
Analyze centrally in secure cloud

Host data locally
Analyze data remotely and collate results

User — Data transmission — Data Visiting - - - ► Results sent back to user

**Figure 1. Data sharing approaches: Central database, secure cloud, and federated**
Central database: Data from multiple sources are pooled in a central database. Researchers download copies of data and analyze them in their own computing environment.
Secure cloud: Data from multiple sources are pooled in a central cloud environment. Researchers remotely visit data and run their analyses in the cloud and download the result.
Federation: Data remain within locally controlled databases and computing environments, which may be cloud environments. Researchers remotely visit data, run their analyses at each site, and receive a local result, which can then be aggregated.

how to incorporate GA4GH standards and frameworks. Federated approaches are justified over alternatives only where data cannot be pooled or transmitted for legal or technical reasons. Success is only likely where data providers have significant resources and a clear mandate to share. Federated approaches can involve different levels of organizational independence and security, with consequences for legal compliance, incentives, and costs. Data and technical standards—key enablers for data sharing generally—are especially vital for federated approaches, ensuring that data are FAIR (findable, accessible, interoperable, and re-useable) so as to enable analysis at scale.[6] Standard-setting bodies like the GA4GH are needed to bring together networks of independent data providers to drive adoption of these standards.

We recommend a federated system to connect national genomics initiatives into a global precision medicine resource. Connecting these resources would provide an opportunity for research on an unprecedented scale. A federated approach is necessary in this context. These initiatives face important security, sovereignty, and trust concerns that militate against pooling data in centralized environments. National initiatives are increasingly integrated with healthcare systems, which

tend to impose stricter rules around confidentiality and secondary use for research (though this depends very much on context). The sheer size of population-scale genomic databases makes them technically difficult to manage and transfer. Nations also expect their investments in large-scale genomic medicine initiatives to serve (competitive) national scientific, health, and wealth goals, with international research agendas being secondary. In light of all these concerns, trust across diverse countries and actors can be hard to establish. A federated approach is also feasible for national initiatives, who have the mandate to share and resources to make data and technical infrastructure—following GA4GH standards—available to the research community. This international use case, if successful, can provide a blueprint for expanding federated approaches to rich, real-time genomic data across national networks of hospitals and laboratories.

**Key design choices and trade-offs**
Federated approaches to data sharing allow data providers to preserve control, security, and accountability while (under the right conditions) still enabling data users to run analyses at scale. The level of data provider independence and the level of security varies across federated

approaches, with important implications for legal compliance, incentives, and costs. The following design considerations and trade-offs, drawn in part from experience in artificial intelligence and digital health contexts,[7,8] provide a guide for the genomics and health community.

*Control over data*
Federated data sharing approaches emphasize the independence of the participating data providers. The Oxford English Dictionary (third edition, 2015) defines federation as a "body . . . formed from a number of separate organizations . . . each retaining control of its own internal affairs." A federated approach to data sharing typically means that data providers retain control over their own data, hosted in their own secure computing environment. Data providers also retain control over access management, i.e., who can access the data, for what purposes, and under what conditions. Greater control is meant to give data providers the confidence to make richer datasets available to a broader range of users, assuming they have the mandate and resources to do so.[5] The degree of individual organizational independence and control varies across federated approaches. At the most independent and loosely defined end, federation may simply be a group of independent data providers who voluntarily adopt a basic set of data

and technical standards. In this approach, there is no global data access committee, and data providers can independently establish their own data access policies. This approach is lightweight for data providers, but it requires data users to make separate access applications for each database and to navigate different access criteria. Although users face more paperwork, they are still able to access and analyze multiple databases separately and then integrate the results.

In more coordinated models of federation, data providers actively collaborate to align data standards and streamline user access. They may even agree to common access rules or to coordinate their access processes through a central data access portal or committee. Sharing sovereignty constrains independence over access management, though data providers still maintain direct control over data. This gives them greater flexibility to withdraw (certain kinds of) access at a later time, if conditions become less favorable. Users benefit from being able to access multiple resources with a single application and to trust their analyses will run reliably in different environments on interoperable datasets.

### Data utility

On the one hand, federated approaches can enhance data utility. They provide a means to combine datasets into a virtual cohort, enabling analyses on datasets of larger scale and statistical power. Because data providers keep tight control over their datasets, they may be more willing and able to share richer, more routinely updated data. De-identification does not need to be as rigorous, as data are not disclosed, preserving utility. On the other hand, the utility of the datasets depends on the adoption of data and technical standards by data providers who require significant resources and expertise. Some data quality issues like record de-duplication can only be addressed collaboratively across data providers. This may be done securely through privacy-preserving record linkage. Users with limited access to data are unable to assess data quality or compare data across sources, exacerbating general data science challenges. They are more reliant on data providers to assist with data curation, analysis, and interpretation. Pooling and direct exchange of data has

long been a catalyst for the standardization of data elements, models, and quality. With no central repository to foster comparison, a federation of independent data providers may need compensating measures to actively drive standardization, such as standard-setting bodies, certifications, or trusted third-party curation services. These challenges can be facilitated by APIs (application programming interfaces) and containers. APIs are interfaces that allow users to query databases even with different underlying data formats. Containers are tools that bundle together software pipelines and their dependencies so they can run reliably in different computing environments.

### Security

In federated data sharing models, each data provider grants authorized users remote access to data in its own computing environment. Access may be direct or indirect.[9] Users granted direct access may analyze each database separately, taking only summary statistics with them when they leave. This limits copying and transmission of data, reducing security risks and allowing continuous monitoring of user activity. The workflow is similar to contexts in which data are pooled centrally, in which users still need to segregate datasets for analytical reasons (e.g., applying different covariates and making independent estimates of significance). For even greater security, users can be limited to indirect access to data. Data remain hidden at all times behind secure firewalls. Users submit algorithms or queries, which are vetted and executed by the data provider, who returns summary or performance statistics.[4] Federated analysis means running the same analysis across multiple hidden databases. This has been demonstrated in artificial intelligence contexts, where models are trained across hospitals[10] or personal smart phones. Only in an idealized vision is federated analysis perfectly seamless for users; data providers may very well insist on their independence to control access to their own data and computing environments. Ultimately, greater data security has tradeoffs. It constrains users' ability to interact with data. Data and technical standards become all-important to

ensure interoperability. Most importantly, the significant costs of both standardization and security fall to the data providers.

Federated data sharing models also introduce new security risks. Data providers face IT security risks when external users, or their software, are introduced into local computing environments. These risks can be alleviated through careful monitoring of user activity and airlocks to control introduction of external software (at additional cost). Federated approaches can also create security risks for users, who expose their research questions or code to a network of data providers. Where risks to users' queries and code are serious, they can be reduced through encryption and secure computing approaches in which data providers execute hidden code.[7]

### Legal compliance and ethics

Federated approaches can alleviate legal and ethical concerns raised by data sharing, though they are not a panacea. The European Union General Data Protection Regulation 2016/679 (GDPR) has set a global standard for robust protection of personal data, which includes mandating limitations on international transfers of personal data outside the EU/EEA. It has also triggered a strong shift toward federated approaches for large scientific data infrastructure, in projects like the European Genome-Phenome Archive, European Open Science Cloud, the European 1+ Million Genomes Initiative, and the European Health Data Space. Secure local data hosting can improve accountability, trust, and individuals' ability to exercise rights like withdrawal of consent to further use or sharing of their data. Robust safeguards provide strong assurances of data protection, even when data are accessed by international researchers. International access within a European data center is still an international transfer, however. Clear legal pathways and privacy-enhancing technologies must be further developed before access can be extended outside Europe.[11] Even where data do not move, appropriate informed consent and ongoing transparency are still generally required for data sharing. Data subjects need to know who is accessing data and for what purposes. Research ethics oversight may also be a greater challenge for federated approaches than alternatives,

as data are analyzed across many different institutions and countries. To address this challenge, the GA4GH Ethics Review Equivalency Policy promotes international standards for ethics review, alongside cross-border coordination and recognition mechanisms.[1]

### Incentives

A lack of incentives to provide data is a well-known barrier to data sharing. While federated approaches do not resolve this barrier, they do give data providers increased control and security, which may increase their willingness to share. Ongoing control may also mean data providers have more leverage to negotiate active collaboration, appropriate scientific recognition, or a share in commercial outputs. More conditions and transaction costs, however, discourage re-use of data, especially as they stack up across data providers. Indirect benefits to data providers include opportunities to develop local capacity and expertise in data infrastructure, management, and analysis. Ultimately, however, incentives must continue to be addressed through broader policy initiatives, investment in infrastructure, and cultural change.

### Sustainability

The most important consideration for data providers considering a federated approach is cost. Data providers incur significant security, data management, and computing costs, including those related to adopting and maintaining standards. These costs are likely to be duplicated across data providers and thus higher overall in comparison to central databases. Federated approaches do spread these costs more evenly across data providers. One way to mitigate expense is through optimal network design. An international federation of genomic databases is enabled by pooling data on a national level. National pooling may raise fewer legal and trust issues, while also providing efficiencies.

### Enabling standards

A key challenge for federated approaches is driving the adoption and maintenance of data and technical standards across numerous, independent organizations. Relying on voluntary adoption of community guidelines is likely to be too weak. Establishing formal partnership agreements could be too strong. The GA4GH, as an open standards-setting body, provides a middle way. It offers a flexible and participatory model to drive the international adoption of consensus standards, collaborating with a network of Driver Projects and member organizations across the global genomics community.

The GA4GH develops and endorses data and technical standards that can be used to enable data sharing generally and federated approaches specifically (see Rehm et al. in this issue for details on these standards[1]). Data and metadata standards are key enablers for any discovery and re-use of data. Standard file formats provide standard structures for genomic data. The Phenotype Ontology provides a semantic ontology for expressing phenotypic data. Federated approaches additionally require technical standards to ensure the interoperability of distributed databases and computing environments. The GA4GH Beacon and Data Connect APIs allow researchers to find individuals with relevant genotypes or phenotypes in a database. Search interfaces can accept structured queries as input and release structured search results as output. Federated search is where users submit a single query that is run on and answered by multiple, independent databases, even where underlying structures differ. Each organization can determine the specificity of the search results (e.g., a simple yes/no, summary statistics, minimal health information associated with the variants) and its own access controls and security safeguards. Federated search has already been successfully demonstrated with GA4GH APIs.[12]

Authentication and authorization standards are needed to coordinate user access to multiple databases. OAuth 2.0 and OpenID Connect are useful tools to assist data providers in confirming the user seeking access is the person who has received approval to do so. Even where data providers retain independent control over access decisions, they may agree to coordinate user authentication protocols. CanDIG, a GA4GH Driver Project, uses an authentication scheme based on OpenID Connect, where each data provider authenticates the identity of its own employees, and that authentication is in turn accepted by the other participating nodes.[13] Each data provider continues to make its own authorization decisions based on local policy. Even so, federated approaches are facilitated where data providers express their local data access and use credentials in a standard way. GA4GH Passports build on authentication standards to allow data providers to confirm a user has standard credentials.[14] The Data Use Ontology (DUO) allows data providers to ensure access requests match to standard data use conditions.[15] Federated analysis in particular requires interoperability between computing environments, because workflows are executed on behalf of data users on hidden databases. Federated analysis can be assisted by the GA4GH Cloud APIs, interfaces that allow users to look up data and tools and to execute portable workflows, driving larger-scale and more powerful analyses. The GA4GH Federated Analysis Systems Project (FASP) brings all these pieces together into end-to-end test scenarios, aiming to simulate how a researcher would search, access, and analyze genomic data across a network of real-world projects.[1]

### Conclusion

Federated approaches to data sharing are flexible, involving design choices about data provider independence and secure access mechanisms. These choices influence data accessibility, data utility, legal compliance, and cost. The GA4GH encourages federated approaches where data providers have the will and resources to share but where data cannot flow because of legal, technical, or institutional policy reasons. Federated approaches come with costs and limitations, but they also provide opportunities to improve privacy protection, accessibility, and interoperability. Advancing federated approaches in genomics will also align the field with data sharing practices in digital health and artificial intelligence.

Creative mechanisms are needed to drive adoption of data and technical standards across networks of independent data providers. As a standards-setting body, the GA4GH is uniquely positioned to assist the genomics community to meet these challenges and bring the vision of a federated approach to genomics and human biomedical data sharing into reality, so as to realize the right of everyone to benefit from the progress of science.

## SUPPLEMENTAL INFORMATION

## WEB RESOURCES

European Genome-Phenome Archive, Federated EGA, https://ega-archive.org/federated

European Commission, European 1+ Million Genomes Initiative, https://digital-strategy.ec.europa.eu/en/news/eu-countries-will-cooperate-linking-genomic-databases-across-borders

European Commission, European Health Data Space, https://ec.europa.eu/health/ehealth/dataspace_en

European Open Science Cloud, https://eosc-portal.eu/

Genomics England, Airlock Policy, Version 2.0, https://www.genomicsengland.co.uk/about-gecip/for-gecip-members/documents/

GA4GH Federated Analysis Systems Project (FASP), https://www.ga4gh.org/genomic-data-toolkit/2020-connection-demos/

GA4GH Ethics Review Recognition Policy, https://www.ga4gh.org/wp-content/uploads/GA4GH-Ethics-Review-Recognition-Policy.pdf

Google AI Blog, Federated Learning: Collaborative Machine Learning without Centralized Training Data, https://ai.googleblog.com/2017/04/federated-learning-collaborative.html

IRDIRC, Technology Primer: Overview of Technological Solutions to Support Privacy-Preserving Record Linkage, https://www.irdirc.org/wp-content/uploads/2018/03/PPRL-Technical-Primer-V4-2.pdf

OAuth 2.0 Authorization Framework, https://datatracker.ietf.org/doc/html/rfc6749

OpenID Connect Core 1.0 incorporating errata set 1, https://openid.net/specs/openid-connect-core-1_0.html

## ACKNOWLEDGMENTS

We acknowledge the encouragement of the Global Alliance for Genomics and Health Steering Committee and the input of its members into the concept and content of the paper. We would like to thank Stephanie Li for assistance with preparing the manuscript graphics. A.T. acknowledges funding support from Genome Canada, Genome Quebec, and the Canadian Institutes of Health Research. H.L.R. and A.J.H.P. acknowledge funding under NIH U41HG006834 and U24HG011025. M. Baudis acknowledges funding under the BioMedIT Network project of Swiss Institute of Bioinformatics (SIB) and Swiss Personalized Health Network (SPHN). M.L. acknowledges funding from the CINECA project (H2020 No 825775). M.S.C. acknowledges funding under NIH/NCI U01CA242954, NIH/NHLBI Fellowship 5118777. M. Brudno is a CIFAR Canada AI Chair. P.G. acknowledges funding from CIHR, Genome Canada, Wellcome Trust, and NIH. T.H.N. was funded by the Academy of Finland grant no 319968 and ELIXIR Europe 2019–2023 program.

## AUTHOR CONTRIBUTIONS

A.T.: Conceptualization; Writing (Original Draft). H.L.R., P.G., E.B.: Conceptualization; Supervision; Writing (Review & Editing). A.J.H.P., Y.J., M. Baudis, J.R., A.N., T.H.N., M.L., E.S.D., MF., M. Brudno, and M.S.C.: Writing (Review & Editing)

## DECLARATION OF INTERESTS

M. Brudno holds financial interest in PhenoTips. E.B. is a consultant to Oxford Nanopore Technologies and Dovetail Inc. and a member of the *Cell Genomics* advisory board. H.L.R. is a member of the *Cell Genomics* advisory board. All other authors have no interests to declare.

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
