## [Document S1. Transparent peer review records for Thorogood et al · Cell Genomics]

Cell Genomics, Volume 1

Supplemental information

**International federation of genomic medicine
databases using GA4GH standards**

Adrian Thorogood, Heidi L. Rehm, Peter Goodhand, Angela J.H. Page, Yann Joly, Michael Baudis, Jordi Rambla, Arcadi Navarro, Tommi H. Nyronen, Mikael Linden, Edward S. Dove, Marc Fiume, Michael Brudno, Melissa S. Cline, and Ewan Birney

International Federation of Genomic Medicine Databases Using GA4GH Standards

Adrian Thorogood (1), Heidi L Rehm (2), Peter Goodhand (3)(4), Angela JH Page (3)(5), Yann Joly (6), Michael Baudis (7), Jordi Rambla (8), Arcadi Navarro (9), Tommi H Nyronen (10)(11)(12), Mikael Linden (10)(11), Edward S. Dove (13), Marc Fiume (14), Michael Brudno (15), Melissa S Cline (16), Ewan Birney (17)

Summary

Scientific Editor:	Orli Bahcall
Initial submission:	2/16/2021
Revision received:	6/07/2021
Accepted:	9/14/2021
Rounds of review:	2
Number of reviewers:	3

Referee reports, first round of review

Reviewer #1: The paper lays out the case for federation of data while acknowledging the value of existing data sharing relationships. It sets up a framework for thinking about the variety of data and compute scenarios across the range of loose to deep federation -- a useful paradigm. It offers GA4GH as a standard setting body without actually proposing any standards.

The authors briefly address the issue of one-way sharing or lack of trust between partners, offering primarily the autonomy of the data owner as a solution. In a world full of data breaches, system administrators are loathe to allow access by outsiders, particularly if outside code will be run (as was addressed on p19: ""There are also security issues about allowing external software into local computing environments.""). The paper raises the issue in several places without much in the way of concrete solutions.

The cost of federation falls on the data host, but the benefits typically accrue to others. The paper makes the point but does not really offer a solution. What incentives can be built into federation that would motivate data-holders to allow access?

The paper is a useful summary of ideas that have bandied about for along time. The argument that federation is a viable solution to problem of data silos would be strengthened by concrete examples of protocols that are working. dbGaP is given as an example, but it is notorious for the difficulty faced by those attempting access.

comments on specific points

p8. scalability of "cloud commons" is in doubt. genomic data is huge. who pays for storage and download?

p9. "Deeper levels of federation are possible between trusted groups, who can agree on common data access governance, greater technical interoperability, and benefit-sharing." mentions trust, but does not address how to obtain it.

The paper suggests standardization of metadata, formats and sharing protocols.

This is important. but who imposes backward conversion? and pays for it? e.g., the simple case of conversions between the two latest genome assemblies is already a barrier to many groups. The paper does address API-driven access as a possible solution for federating databases using disparate protocols or formats.

"define stds" -- but does that means abandoning legacy data?

p17. "There are, however, very few laws that categorically prohibit the transmission of personal data outside institutions or across borders. It is rather that certain conditions have to be fulfilled."

China? "but a permit can be obtained to do so" are they in practice, granted?

" Under British Columbia's Public Sector Privacy Act, public bodies in the Canadian province face restrictions on the transfer of personal information outside the country, but can do so with the individual's consent."

Individual consent is impossible to obtain in many cases -- either by original protocol design or the practical barriers for re-contact with patients.

p18 "The GA4GH is focusing on developing standards to make federation a possibility."

It would be good if there were such standards being proposed in this paper.

p19. "While the initial aim of federation is to enable the sharing of data that cannot otherwise be shared, there is a function-creep risk that the availability of federation erodes the willingness of data holders to share through more open approaches."

Interesting idea.

nits:

Paper uses "data" as both singular and plural -- at least once in the same sentence

p7 v.s. > vs.

p17 " Data transfers are rarely legally or technically impossible; rather, they are subject to high compliance or technical burdens mean that data transfer cannot be conducted in a timely and affordable manner."

This sentence is difficult to understand.

Does it mean:

" Data transfers are rarely legally or technically impossible; rather, they are subject to high compliance costs or technical burdens mean that data transfer cannot be conducted in a timely and affordable manner."

?

p19. "Federation retains some incentivizes" > incentives.

Reviewer #2:

The paper seems to me to consist of two parts, which are quite independent.

1. I would like to suggest that GA4GH come up with standard definitions of federation, develop use cases matching the definitions, and map the definitions and use cases to their current and emerging standards. I think GA4GH is well positioned to do this and it would be very valuable to the community, and most welcome. As pointed out in Table 1, the current and emerging GA4GH standards play a very important role in this process. Except for Table 1, I don't see this manuscript as contributing materially to this process.

The GA4GH Federated Analysis Systems Project (FASP) is a step in this direction, as is the NIH Cloud Platform Interoperability (NCPI) Working Group, their definitions, and their use cases.

2. I think the GA4GH recommendations regarding federation (Section II: To Federate or not to Federate) would make a good perspective piece, and would only require a few paragraphs of background about federation. I think that would be an important contribution that GA4GH could make.

In my opinion, 1) requires substantial work and effort by GA4GH before a ms can be prepared, but would be extremely valuable. On the other hand 2) is ready now and is timely.

Here are more detailed comments about the ms.

Page 4: The authors write: "The primary aim of this paper is to define a concrete vision for federation in the context of international genomic data sharing, highlighting both flexibility in the depth of implementation, and the central role technical standard-setting plays in its realization."

Page 5: The authors write: "A second aim of this paper is to clarify the GA4GH position on when and how federation should be pursued."

"Our key argument is that federation is a valuable complement to our existing data sharing methods, not a universal substitute (see Figure 1)."

This are good aims, but I don't feel the paper has delivered on them.

Page 6. The distinction of the authors between data commons and cloud commons is not standard. Data commons can be configured to allow or disable downloads; many data commons support and even require that analysis take place within the cloud-based infrastructure; and, there are very different degrees of protection of when data is analyzed in clouds. The authors are free to choose any names for different categories of data platforms, including "genomic cloud platforms", but they should carefully define them and distinguish them. Also, they need to clarify, that just because data is analyzed within a cloud-based data platform, they are widely different approaches to security, compliance and egress controls.

Page 6. I find Figure 1 confusing and misleading. Many data commons harmonize data (that is analyze data centrally) and then permit further analysis either in a cloud platform or after downloading. Many data commons also support large scale research. Calling the idea of federation "new" ("Federation - new approach for national genomic initiatives") does not seem accurate to me.

Page 6. Referring to data commons and cloud commons as the "Status Quo" also appears misleading to me, given the large amount of innovation and effort being devoted to these platforms. As the authors point out, there has always been a mixture of centralization and federation in analysis in general, and genomic analysis in particular. With the growing capabilities of cloud platforms and the growing sizes of genomics databases, this mixture is continuing to change.

Page 6: The authors write: "A cloud commons provides access not only to data, but also to methods, workflows, and computing resources." If this is the definition the authors will be using in this paper, they should state this.

Page 7. Since it is one of the aims of the paper to revisit federation and the GA4GH's role in providing supporting standards and recommendations for it, this material needs to be more carefully organized and presented, with definitions given for loose vs deep federation and vertical vs horizontal federation.

Presumably, this is the definition of "Loose federation": "Researchers seeking to integrate an analysis across multiple databases providing only remote access can still do so by requesting access to each resource individually. They can instruct software machines to run an analysis on each database they are authorized to access, and then pool the summary statistics. This "loose" form of federation essentially enables a form of individual participant data (IPD) meta-analysis..." (Page 7). If so, they should state this.

Page 7. Last paragraph. The GA4GH has done significant work on several different standards related to federation. None of that is reflected in this paragraph, with sentences such as: "datasets are hosted behind secure firewalls at physically and organizationally distinct locations across a network."

Page 8. The authors should either define deep federation, or if they prefer place loose and deep federation on a spectrum in a table or figure, and place the characteristics referenced in the appropriate place. For example, "A unique characteristic of this deeper model of federation is that users are granted algorithmic—not direct—access to data. (Page 8)" Here again the concept is ill-defined and language is imprecise. Providing users algorithmic vs direct access to data is a very old concept, described in the references, cited and is present in multiple models of federation and distribution.

Page 9. The authors write: "Federation and centralization can, and often do, co-exist in the same network of databases. Nodes in a network may themselves be made up of centralized databases pooled from multiple organizations, sometimes referred to as a hybrid network.¹⁰ It is even possible to federate already existing centralized databases or clouds hosting organizational, national, or regional data. The question of whether to federate or centralize can be answered differently at different layers within a network."

...

"Additionally, networks can choose to federate some aspects of data processing but not others. For example, a network can federate search, but still permit physical pooling for analysis. Different activities may, however, be practically interdependent: data discovered through a federated search will have more utility if that search is complemented by federated analysis."

I agree with this analysis, but this points out the need to more carefully and thoughtfully structure the discussion on pages 4-9 of the ms.

The authors should make clear that all the approaches to federation are standard and not new, and the (important) contribution of GA4GH is to develop community consensus and put in place standards that can contribute to different approaches to federation as Table 1 illustrates.

Pages 10-12. Figure 2. This figure needs work. Presumably "Data Visiting" means the user logs into each system separately, analyzes the data, downloads the data, and analyzes the downloaded data. Presumably, there is some degree of standardization / commonality horizontally across blue boxes with the same name. I'm not sure I would call the different sub-figures "models," since that seems to imply that there are only three different models of federation, vs all the differences that arise combinatorially, as different horizontal and vertical federation choices are made.

Pages 15-17 - Section II: To Federate or not to Federate

This is a nice summary of whether to federate or not to federate and would make a good perspective piece.

Reviewer #3: This paper discusses the use of federation and for large scale joint data analysis. It posits that clinical genomic testing will become an increasingly large and important source of genomic data in the future, which is likely correct. Given the careful natures in which clinical data are regulated, the need to federated data analyses - already very important in genomic analysis - will become increasingly important to harness the ability to "learn something from every patient".

A real strength of this paper is a discussion of when to use federation for data analysis. It is noted appropriately that federation should be adopted universally, but only when exchange of data is not otherwise possible. An important downside of federation that could be strengthened is that federation can make data look artificially comparable through the process of standardization. For instance, variability in how a data element was collected can be obscured when mapping to a common vocabulary. In addition, federation often proceeds via meta-analysis and sometimes the row-level data is not visible. This further obscured the variability in the underlying data.

Another consideration for data sharing: Given that the context of this paper largely is around the massive growth of clinical genomic datasets, another avenue for data sharing that could be discussed in the paper is via the patient/participant themselves. Technologies like patient portal-based downloads using common standards such as FHIR allow for patient-driven research. This is being seen in some COVID networks, All of Us, and in the past for BRCA. EHR certification for common data will make this increasingly common, and the Sync4Genes effort is a step in a common exchange format. In addition, networks like Health Information Exchanges, which currently rely primarily on purely clinical data, could expand to share (clinical) genomic data for clinical uses. One could imagine HIEs could adopt research use cases as well.

Another minor addition to the paper would be to note that HIPAA allows for sharing of identified data as part of treatment, payment, and operations already, without consent. Such sharing could result in large, shared datasets of genomic data at certain clinical or payment-oriented entities within the US. Notably, however, research use cases are not allowed without de-identification, patient consent, or an IRB-approved research study that could exempt the study's need for consent.

Another challenge to recognize with federation, primarily, over centralized research data is the need to invoke a protocol for deduplication. This can be challenging in any circumstance, but it is harder in federated datasets. There is great work with privacy preserving data linkage processes, however, that can enable deduplication and linkage of otherwise-firewalled data sets.

Minor feedback:

Summary: define "GA4GH" at first use

Page 7, paragraph three. "deeper levels of federation or possible between trusted groups" -- one would argue one benefit of federation is that you can have a lower bar of trust between entities. One just needs to agree on common data, Interoperability, and access standards in many cases, and federated entities can employ their own trust verification processes.

Figure 2 - Could this figure be adapted to show the role of common data standards as well? For instance, you could use different colors or different shapes to represent the change from raw data to a curated data set that could be commonly searched and analyzed.

In Table 1, there are a handful of abbreviations and references that could be defined.

Another small point was the use of the word "physical" for the movement of data and where it resides.

Author response to the first round of review

Response to Reviewers

Editor Comments

We invite you to revise your manuscript in response to these comments.

In addition, we feel the manuscript would benefit from streamlining the text to focus on the main messages on defining federation models, the need and applications for these in genomic medicine, GA4GH contributions and role, and when to federate or not. An alternative direction would be to refocus more as a GA4GH position paper on federation. I will be glad to discuss how to best focus and present this work, in context of this manuscript, and as part of the special issue.

We are particularly interested to see additional work and discussion included in response to Referee #2's comment #1 on defining federation.

We also recommend presenting the manuscript in either our Perspective or Commentary format, and I would be glad to discuss your preferences.

I will be delighted to discuss these and additional editing suggestions and your plans for revising, to support you in the optimal presentation for this exciting and impactful publication.

We have significantly streamlined the text and refocused it as a GA4GH position paper. The introduction now briefly defines federation and provides the basic rationale for its application in genomics. The body of paper focuses on trade-offs, that are highly dependent on how the flexible concept is practically implemented. We highlight relevant GA4GH standards throughout. We believe given the current focus, and reduced length (around 3000 words), the paper could be effectively presented as a Commentary.

Reviewer #1

The paper lays out the case for federation of data while acknowledging the value of existing data sharing relationships. It sets up a framework for thinking about the variety of data and compute scenarios across the range of loose to deep federation -- a useful paradigm. It offers GA4GH as a standard setting body without actually proposing any standards.

The authors briefly address the issue of one-way sharing or lack of trust between partners, offering primarily the autonomy of the data owner as a solution. In a world full of data breaches, system administrators are loathe to allow access by outsiders, particularly if outside code will be run (as was addressed on p19: ""There are also security issues about allowing external software into local computing environments.""). The paper raises the issue in several places without much in the way of concrete solutions.

In the section “security” we clarify that there are security risks for data provider’s IT environments, but now add that these can be addressed through monitoring and airlocks.

The cost of federation falls on the data host, but the benefits typically accrue to others. The paper makes the point but does not really offer a solution. What incentives can be built into federation that would motivate data-holders to allow access?

In the section “incentives and sustainability” we now explicitly clarify that federation alone cannot address the incentives issue for data sharing. This is why it is most appropriate in contexts like national genomics initiatives where the organization has the mandate and resources to make data available. We do mention some of the modest benefits that come with greater ongoing control.

The paper is a useful summary of ideas that have bandied about for along time. The argument that federation is a viable solution to problem of data silos would be strengthened by concrete examples of protocols that are working. dbGaP is given as an example, but it is notorious for the difficulty faced by those attempting access.

We are not currently able to identify models of federation with all of the characteristics we describe.

comments on specific points

p8. scalability of "cloud commons" is in doubt. genomic data is huge. who pays for storage and download?

To avoid confusion, we now refer to secure clouds instead of cloud commons. We present these as clouds that do not allow download. Still, hosting and user analyses can be costly. We do not aim to directly compare federation with secure clouds in this limited space. We now highlight in the section on “incentives and sustainability” that an advantage of federation is that costs may be naturally spread out more evenly across organizations.

p9. "Deeper levels of federation are possible between trusted groups, who can agree on common data access governance, greater technical interoperability, and benefit-sharing." mentions trust, but does not address how to obtain it.

Our general point is that federation does not depend in the first instance on this deep level of trust, and also that enabling federated analysis where users only have algorithmic access is both technically and organizationally challenging to achieve. However we do now mention that one way trust is achieved is through repeated interactions. (See section “Control over data”)

The paper suggests standardization of metadata, formats and sharing protocols. This is important. but who imposes backward conversion? and pays for it? e.g., the simple case of conversions between the two latest genome assemblies is already a barrier to many groups.

The paper does address API-driven access as a possible solution for federating databases using disparate protocols or formats.

This point is covered in our general section on incentives and sustainability. Federation alone does not address incentives issues. In our use case, it is clearly the nations / national initiatives who pay.

"define stds" -- but does that means abandoning legacy data?

Standards can be applied equally to legacy data, so we have not addressed this comment in text.

p17. "There are, however, very few laws that categorically prohibit the transmission of personal data outside institutions or across borders. It is rather that certain conditions have to be fulfilled."

The section on what constitutes a true legal barrier has largely been removed. There is not a short section nuancing the consequences of federation for ethics and data protection compliance.

China? "but a permit can be obtained to do so" are they in practice, granted?

See above.

" Under British Columbia's Public Sector Privacy Act, public bodies in the Canadian province face restrictions on the transfer of personal information outside the country, but can do so with the individual's consent."

Individual consent is impossible to obtain in many cases -- either by original protocol design or the practical barriers for re-contact with patients.

The short section on data protection and ethics highlights that federation may not get around consent requirements.

p18 "The GA4GH is focusing on developing standards to make federation a possibility."

It would be good if there were such standards being proposed in this paper.

These are now highlighted in the main text, instead of in a separate table.

p19. "While the initial aim of federation is to enable the sharing of data that cannot otherwise be shared, there is a function-creep risk that the availability of federation erodes the willingness of data holders to share through more open approaches."

Interesting idea.

nits:

Paper uses "data" as both singular and plural -- at least once in the same sentence

We have reviewed this for consistency.

p7 v.s. > vs.

No longer in the text.

p17 " Data transfers are rarely legally or technically impossible; rather, they are subject to high compliance or technical burdens mean that data transfer cannot be conducted in a timely and affordable manner." This sentence is difficult to understand. Does it mean: " Data transfers are rarely legally or technically impossible; rather, they are subject to high compliance costs or technical burdens mean that data transfer cannot be conducted in a timely and affordable manner."?

No longer in the text.

p19. "Federation retains some incentivizes" > incentives.

Addressed.

Reviewer #2:

The paper seems to me to consist of two parts, which are quite independent.

1. I would like to suggest that GA4GH come up with standard definitions of federation, develop use cases matching the definitions, and map the definitions and use cases to their current and emerging standards. I think GA4GH is well positioned to do this and it would be very valuable to the community, and most welcome. As pointed out in Table 1, the current and emerging GA4GH standards play a very important role in this process. Except for Table 1, I don't see this manuscript as contributing materially to this process.

This comment has prompted much of our restructuring. We define federation in the introduction. With streamlining, we do not develop use cases, but instead we discuss different practical implementations and the associated trade-offs. Instead of Table 1, we include a section describing GA4GH standards needed for different implementations, and highlighting GA4GH's community leadership role driving standards adoption.

The GA4GH Federated Analysis Systems Project (FASP) is a step in this direction, as is the NIH Cloud Platform Interoperability (NCPI) Working Group, their definitions, and their use cases.

While we mention FASP, we do not described detailed use cases in this shorter version, but rather practical implementations.

2. I think the GA4GH recommendations regarding federation (Section II: To Federate or not to Federate) would make a good perspective piece, and would only require a few paragraphs of background about federation. I think that would be an important contribution that GA4GH could make. In my opinion, 1) requires substantial work and effort by GA4GH before a ms can be prepared, but would be extremely valuable. On the other hand 2) is ready now and is timely.

See our general response to the editor.

Here are more detailed comments about the ms.

Page 4: The authors write: "The primary aim of this paper is to define a concrete vision for federation in the context of international genomic data sharing, highlighting both flexibility in the depth of implementation, and the central role technical standard-setting plays in its realization."

Page 5: The authors write: "A second aim of this paper is to clarify the GA4GH position on when and how federation should be pursued."

"Our key argument is that federation is a valuable complement to our existing data sharing methods, not a universal substitute (see Figure 1)."

This are good aims, but I don't feel the paper has delivered on them.

See our general response to the editor.

Page 6. The distinction of the authors between data commons and cloud commons is not standard. Data commons can be configured to allow or disable downloads; many data commons support and even require that analysis take place within the cloud-based infrastructure; and, there are very different degrees of protection of when data is analyzed in clouds. The authors are free to choose any names for different categories of data platforms, including "genomic cloud platforms", but they should carefully define them and distinguish them. Also, they need to clarify, that just because data is analyzed within a cloud-based data platform, they are widely different approaches to security, compliance and egress controls.

We now refer to data commons as the general category, that can have diverse designs such as central databases with central databases stored in secure clouds. The point on cloud diversity was well articulated by the reviewer and was added in the introduction.

Page 6. I find Figure 1 confusing and misleading. Many data commons harmonize data (that is analyze data centrally) and then permit further analysis either in a cloud platform or after downloading. Many data commons also support large scale research. Calling the idea of federation "new" ("Federation - new approach for national genomic initiatives") does not seem accurate to me.

First we have updated the titles and descriptions in the Figure to address this comment. Second, we avoid the word "New" when describing federation.

Page 6. Referring to data commons and cloud commons as the "Status Quo" also appears misleading to me, given the large amount of innovation and effort being devoted to these platforms. As the authors point out, there has always been a mixture of centralization and federation in analysis in general, and genomic analysis in particular. With the growing capabilities of cloud platforms and the growing sizes of genomics databases, this mixture is continuing to change.

We avoid the term status quo, highlighting the flexibility and evolution of all models.

Page 6: The authors write: "A cloud commons provides access not only to data, but also to methods, workflows, and computing resources." If this is the definition the authors will be using in this paper, they should state this.

We have nuanced our description of cloud commons in the introduction.

Page 7. Since it is one of the aims of the paper to revisit federation and the GA4GH's role in providing supporting standards and recommendations for it, this material needs to be more carefully organized and presented, with definitions given for loose vs deep federation and vertical vs horizontal federation.

We have removed the terms loose and deep federation. Instead we clarify that federation can involve different levels of organizational independence and security across a spectrum. And in turn different standards are needed.

Presumably, this is the definition of "Loose federation": "Researchers seeking to integrate an analysis across multiple databases providing only remote access can still do so by requesting access to each resource individually. They can instruct software machines to run an analysis on each database they are authorized to access, and then pool the summary statistics. This "loose" form of federation essentially enables a form of individual participant data (IPD) meta-analysis..." (Page 7). If so, they should state this.

We have removed the terms loose and deep federation. See rationale above.

Page 7. Last paragraph. The GA4GH has done significant work on several different standards related to federation. None of that is reflected in this paragraph, with sentences such as: "datasets are hosted behind secure firewalls at physically and organizationally distinct locations across a network."

We have made more systematic mention of GA4GH standards throughout.

Page 8. The authors should either define deep federation, or if they prefer place loose and deep federation on a spectrum in a table or figure, and place the characteristics referenced in the appropriate place. For example, "A unique characteristic of this deeper model of federation is that users are granted algorithmic—not direct—access to data. (Page 8)" Here again the concept is ill-defined and language is imprecise. Providing users algorithmic vs direct access to data is a very old concept, described in the references, cited and is present in multiple models of federation and distribution.

We have removed the terms loose and deep federation. See rationale above.

Page 9. The authors write: "Federation and centralization can, and often do, co-exist in the same network of databases. Nodes in a network may themselves be made up of centralized databases pooled from multiple organizations, sometimes referred to as a hybrid network.¹⁰ It is even possible to federate already existing centralized databases or clouds hosting organizational, national, or regional data. The question of whether to federate or centralize can be answered differently at different layers within a network." ... "Additionally, networks can choose to federate some aspects of data processing but not others. For example, a network can federate search, but still permit physical pooling for analysis. Different activities may, however, be practically interdependent: data discovered through a federated search will have more utility if that search is complemented by federated analysis." I agree with this analysis, but this points out the need to more carefully and thoughtfully structure the discussion on pages 4-9 of the ms.

We now describe briefly the importance of "optimal" centralization as part of our section on "incentives and sustainability".

We have removed a lot of the content on flexibility of implementation, but we still make brief mention in the "security" section.

The authors should make clear that all the approaches to federation are standard and not new, and the (important) contribution of GA4GH is to develop community consensus and put in place standards that can contribute to different approaches to federation as Table 1 illustrates.

As mentioned above, we no longer refer to federation as "new", but rather as an alternative. Our final sections highlight the importance of both GA4GH leadership and standards that support flexible implementations.

Pages 10-12. Figure 2. This figure needs work. Presumably "Data Visiting" means the user logs into each system separately, analyzes the data, downloads the data, and analyzes the downloaded data. Presumably, there is some degree of standardization / commonality horizontally across blue boxes with the same name. I'm not sure I would call the different sub-figures "models," since that seems to imply that there are only three different models of federation, vs all the differences that arise combinatorially, as different horizontal and vertical federation choices are made.

Figure 2 has been removed. The original figure conflated the concept of access control with the concept of secure access. These concepts are now clarified in the text (see sections "control" and "security").

Pages 15-17 - Section II: To Federate or not to Federate

This is a nice summary of whether to federate or not to federate and would make a good perspective piece.

Reviewer #3:

This paper discusses the use of federation and for large scale joint data analysis. It posits that clinical genomic testing will become an increasingly large and important source of genomic data in the future, which is likely correct. Given the careful natures in which clinical data are regulated, the need to federated data analyses - already very important in genomic analysis - will become increasingly important to harness the ability to "learn something from every patient".

A real strength of this paper is a discussion of when to use federation for data analysis. It is noted appropriately that federation should [not] be adopted universally, but only when exchange of data is not otherwise possible.

An important downside of federation that could be strengthened is that federation can make data look artificially comparable through the process of standardization. For instance, variability in how a data element was collected can be obscured when mapping to a common vocabulary. In addition, federation often proceeds via meta-analysis and sometimes the row-level data is not visible. This further obscured the variability in the underlying data.

This data quality issue is an important point for data sharing generally. It is a bit beyond our focus on the comparative advantages/disadvantages of federation. In our "data utility" section we now mention that federation's limited access to data may exacerbate general data science challenges.

Another consideration for data sharing: Given that the context of this paper largely is around the massive growth of clinical genomic datasets, another avenue for data sharing that could be discussed in the paper is via the patient/participant themselves. Technologies like patient portal-based downloads using common standards such as FHIR allow for patient-driven research. This is being seen in some COVID networks, All of Us, and in the past for BRCA. EHR certification for common data will make this increasingly common, and the Sync4Genes effort is a step in a common exchange format.

This is out of scope, as our discussion is agnostic to how genomic projects obtain their data.

In addition, networks like Health Information Exchanges, which currently rely primarily on purely clinical data, could expand to share (clinical) genomic data for clinical uses. One could imagine HIEs could adopt research use cases as well.

We already mention the opportunities for genomic models to align with healthcare ones in our conclusion.

Another minor addition to the paper would be to note that HIPAA allows for sharing of identified data as part of treatment, payment, and operations already, without consent. Such sharing could result in large, shared datasets of genomic data at certain clinical or payment-oriented entities within the US. Notably, however, research use cases are not allowed without de-identification, patient consent, or an IRB-approved research study that could exempt the study's need for consent.

The consent/ethics review aspects here are addressed in our shorter section on data protection and ethics. For simplicity we focus mainly on research re-use and we do not address all the regulatory complexities of different re-use purposes.

Another challenge to recognize with federation, primarily, over centralized research data is the need to invoke a protocol for deduplication. This can be challenging in any circumstance, but it is harder in federated datasets. There is great work with privacy preserving data linkage processes, however, that can enable deduplication and linkage of otherwise-firewalled data sets.

We now mention this in the data utility section.

Minor feedback:

Summary: define "GA4GH" at first use

Done.

Page 7, paragraph three. "deeper levels of federation or possible between trusted groups" -- one would argue one benefit of federation is that you can have a lower bar of trust between entities. One just

needs to agree on common data, Interoperability, and access standards in many cases, and federated entities can employ their own trust verification processes.

Mentioned in data control section.

Figure 2 - Could this figure be adapted to show the role of common data standards as well? For instance, you could use different colors or different shapes to represent the change from raw data to a curated data set that could be commonly searched and analyzed.

Figure 2 was removed - see rationale above.

In Table 1, there are a handful of abbreviations and references that could be defined.

Table 1 was removed. We have paid attention to abbreviations in the text.

Another small point was the use of the word "physical" for the movement of data and where it resides.

We have removed this term and now refer to "data transmission" for clarity.

Referee reports, second round of review

Reviewer #1: The paper has become a general but somewhat vague endorsement of federation as a concept. Figure 1 shows several data models, but no caption to describe the differences among the three data-access models. The text of the paper dives right into touting the advantages of federation without a concise description of the alternatives.

The paper offers a summary of the considerations for adopting a federated approach to data sharing, but does not really offer very much that has not been said before.

comments on specific points

p4 of 11. Figure 1. Blue arrows could be labelled "Raw data transmission"? In the center example, one presumes that some summary data are transmitted back to the user? The absence of a "Data transmission" blue arrow there looks like no data at all are returned. Also, in the center section, is the dotted line on the yellow arrow significant? Is there is difference btw the data barrel in the left panel and the cloud in the center?

p7-8 of 11.

"The GA4GH File Formats provide standard structures for genomic data. The GA4GH Phenotype Ontology provides a semantic ontology for expressing phenotypic data."

I do not think it is appropriate to label file formats as "GA4GH" when most file formats in use in genomics

were developed and in common use well before the inception of GA4GH. Similarly, HPO, at least, predates GA4GH.

nits:

p5 of 11.

"This gives them greater flexibility to withdraw (certain kinds) of access at a later time,"

misplaced parens. should be:

This gives them greater flexibility to withdraw (certain kinds of) access at a later time,

p7 of 11.

"This general problem that cannot be resolved simply by adopting federation."

is missing a verb:

This is a general problem that cannot be resolved simply by adopting federation.

OR

This general problem cannot be resolved simply by adopting federation.

?

p8-9 of 11.

"While we focus on connecting national genomics initiatives, they considerations may also inform federation strategies in other contexts."

"these considerations" maybe ??

Reviewer #2:

Comments on Revised Version

I think the revised paper address the concerns raised below well and the community would benefit from seeing it published.

Author response to the second round of review

Response to Editor and Reviewers:

Regarding the helpful editorial text and comments in the manuscript document, we have accepted most of your in text edits and elaborated where requested.

First we address the general comments, that include continuing to question the clarity in the focus and messaging, and not having provided new or clear arguments that federation is a workable situation.

Clarity and Novelty

We have sharpened the focus of the Commentary. We have clarified the arguments throughout. The introduction has been reordered and now includes a short summary of our key points. We also accept your suggestion to focus the summary on a short definition, conditions where federation is appropriate, and the suitability of federation for connecting national initiatives. We have also simplified terminology and provided clear in text definitions of the remaining key terms (e.g., data provider, federated approaches to data sharing; federated search, federated analysis). We clarify the scope of the analysis applies to all genomic and health-related databases, while highlighting that a federated approach to connect national genomics initiatives is a convincing use case, one that we clearly recommend. We have clarified the standards section, distinguishing when standards are generic for data sharing, or are more specifically important for federation. We use the Marker Paper as a general reference.

Rather than novelty, the paper aims to provide a concise, practical summary. It aims not to present new arguments per se, but to align community views and promote a shared understanding of practical aspects of implementation and the implications of different federated approaches. If anything, we agree with the reviewer that federated approaches are hard to achieve, and caution they should only be adopted where necessary, with a clear implementation plan, and where there are mandates and resources to share in this way.

Authorship

One important comment was on authorship, where you requested clarification as to whether our reference to “on behalf of the GA4GH Steering Committee” meant that this committee was the author, or one of the authors. We have decided to remove “on behalf of the GA4GH Steering Committee” from after the authorship list. This was initially intended to indicate endorsement not authorship. We see now that this was not entirely clear in our first submission. We feel it would not be appropriate at this late stage to introduce the broader group as authors. Even the Marker Paper was authored by a list of individuals rather than by all members of the Steering Committee, so it would be strange to adopt this approach in the context of this practical guidance Commentary. We do acknowledge the SC for contributing to the conceptualization of the article, as this began as a SC discussion and mandate.

Terms and Definitions

You request clarity on terms and definitions. We have removed unnecessary or vague terms (e.g., data commons, organization), and used other terms consistently (federated approaches (to data sharing); data provider). Important terms we define clearly in text when first used (e.g., federated approaches, data provider, federated search, federated analysis). We have not introduced a definitions section, as we felt this would unnecessarily interrupt the flow of the article. We clarify in the first sentences and with a reference to the Marker Paper that national initiatives are national-scale genomic sequencing initiatives. We now avoid the vague term organization, and refer consistently to data providers and data users (defined in the introduction).

Re-organization

You requested some reorganization to highlight key points. We have revised the order of paragraphs in the introduction as suggested. We have tried to bring out the message that federated approaches are useful, necessary and feasible for connecting national genomics initiatives, as a blueprint for national networks. We now introduce the FAIR principles and their relation to standards early in the paper. We have split up incentives/sustainability into 2 sections, and distilled the messages there. We have included mention of DUO in federated contexts. We cite the CanDIG paper, but not the Passports paper (which we assume will be referred to in the Marker paper).

Box 1 and Figure 1

There were also some comments concerning Box 1 and Figure 1. We have added a descriptive caption for Figure 1, and we have also updated the terms and images as suggested by the reviewer. We have removed the descriptive Box 1 and integrated the content into the text, as the Box appeared to be missing important context and did not stand alone.

Commentary Format

The formatting has been adjusted to meet the Commentary format. The summary has been cut and the overall length as well to meet these requirements. References are under 15, and now include refs to companion GA4GH papers. We have introduced website links and a website section at the end to refer to certain initiatives and standards. We have provided the requested companion files and declarations of interest form and manuscript disclosures from all authors.

Response to Reviewers

Our response to reviewer comments are below in italics.

Reviewers' Comments:

Reviewer #1: The paper has become a general but somewhat vague endorsement of federation as a concept. Figure 1 shows several data models, but no caption to describe the differences among the three data-access models. The text of the paper dives right into touting the advantages of federation without a concise description of the alternatives.

We have provided a caption for the Figure, describing the different approaches.

The paper offers a summary of the considerations for adopting a federated approach to data sharing, but does not really offer very much that has not been said before.

Please see our comment above about novelty.

comments on specific points

p4 of 11. Figure 1. Blue arrows could be labelled "Raw data transmission"? In the center example, one presumes that some summary data are transmitted back to the user? The absence of a "Data transmission" blue arrow there looks like no data at all are returned. Also, in the center section, is the dotted line on the yellow arrow significant? Is there is difference btw the data barrel in the left panel and the cloud in the center?

We have added a caption and updated the Figure to address these points.

p7-8 of 11.

"The GA4GH File Formats provide standard structures for genomic data. The GA4GH Phenotype Ontology provides a semantic ontology for expressing phenotypic data."

I do not think it is appropriate to label file formats as "GA4GH" when most file formats in use in genomics were developed and in common use well before the inception of GA4GH. Similarly, HPO, at least, predates GA4GH.

We now clarify the provenance of the standards.

nits:

p5 of 11.

"This gives them greater flexibility to withdraw (certain kinds) of access at a later time,"

misplaced parens. should be:

This gives them greater flexibility to withdraw (certain kinds of) access at a later time,

Addressed.

p7 of 11.

"This general problem that cannot be resolved simply by adopting federation."

is missing a verb:

This is a general problem that cannot be resolved simply by adopting federation.

OR

This general problem cannot be resolved simply by adopting federation.

?

Addressed

p8-9 of 11.

"While we focus on connecting national genomics initiatives, they considerations may also inform federation strategies in other contexts."

"these considerations" maybe ??

Addressed.

Reviewer #2:

Comments on Revised Version

I think the revised paper address the concerns raised below well and the community would benefit from seeing it published.

Nothing to address here.